

# The modern spectrum of biopsy-proven renal disease in Chinese diabetic patients—a retrospective descriptive study

Diankun Liu[1], Ting Huang[1], Nan Chen[2], Gang Xu[3], Ping Zhang[4], Yang Luo[5], Yongping Wang[1], Tao Lu[1], Long Wang[1], Mengqi Xiong[1], Jian Geng[6,7] and Sheng Nie[1]

[1] The National Clinical Research Center for Kidney Disease, State Key Laboratory of Organ Failure Research, Nanfang Hospital, Southern Medical University, Guangzhou, China
[2] Ruijin Hospital, Shanghai Jiao Tong University, Shanghai, China
[3] Tongji Hospital, Huazhong University of Science and Technology, Wuhan, China
[4] Sichuan Provincial People's Hospital, Sichuan Academy of Medical Science, Chengdu, China
[5] Beijing Shijitan Hospital, Capital Medical University, Beijing, China
[6] King Medical Diagnostics Center, Guangzhou, China
[7] School of Basic Medical Sciences, Southern Medical University, Guangzhou, China

Corresponding author
Sheng Nie, niesheng0202@126.com

## ABSTRACT

**Background.** Renal biopsies performed in diabetic patients are increasing and becoming more complex. Comprehensive data on modern spectrum of biopsy-proven renal disease in Chinese diabetic patients are lacking.

**Methods.** In a nationwide renal biopsy survey including 71,151 native biopsies from 2004 to 2014, diabetic patients were identified according to the clinical diagnosis from referral records. The clinical data were extracted from referral records and pathological reports.

**Results.** A total of 1,604 diabetic patients, including 61 patients with T1DM, were analyzed in this study. The median age is $51.39 \pm 11.37$ years. Male patients accounted for 58% of the population. We found that only 44.7% of diabetic patients had the isolated pathological diagnosis of diabetic nephropathy (DN), while 49.1% had non-diabetic renal disease (NDRD) alone, and 6.2% had NDRD superimposed on DN. Nephrotic syndrome ($n = 824$, 51.4%) was the most common clinical indication for renal biopsy. Among 887 patients with NDRD, membranous nephropathy ($n = 357$) was the leading diagnosis, followed by IgA nephropathy ($n = 179$). Hypertensive renal disease ($n = 32$), tubulointerstitial nephropathy ($n = 27$) and acute tubular necrosis ($n = 16$) accounted for 3.5%, 2.9%, 1.7% of the NDRD cases respectively. Nearly a half (49.2%) of patients with T1DM had NDRD.

**Discussion.** Over 55% diabetic patients with kidney disease were diagnosed as non-diabetic renal disease, among which MN and IgAN were the most common two pathological types.

## INTRODUCTION

Chronic kidney disease (CKD) is a public health problem all over the world, which has drawn an increased attention in the past 10 years. The overall prevalence of CKD is about 10.8% in China (*Zhang et al., 2012*). It is considered that chronic glomerulonephritis was the leading cause of end-stage renal disease (ESRD) in China before 2010 (*Zuo, Wang & for Beijing Blood Purification Quality Control and Improvement Cente, 2013*), though diabetic nephropathy (DN) has been the predominant etiology in developed countries (*Rocco & Berns, 2012*). With the rapid growth in economy, extension in life expectancy and changes in lifestyle, the prevalence of diabetes mellitus (DM) is progressively increasing in China (*Wild, 2004*). The age-standardized prevalence of diabetes is 9.7% in 2010, which means that nearly 92.4 million adults had DM (*Yang et al., 2010*). Recent studies have evaluated the trends in CKD related to DM and glomerulonephritis in China. Since 2011, the percentage of CKD related to DM exceeded that related to glomerulonephritis (*Zhang et al., 2016*). Therefore, the burden of CKD related to DM exerted on social economy and public health issues cannot be ignored.

The reported prevalence of DN varies widely worldwide. It is generally thought that the clinical onset of DN is characterized by the presence of micro-albuminuria or reduction in estimated glomerular filtration rate (eGFR) (*Kramer et al., 2003*; *Pavkov et al., 2012*; *Shimizu et al., 2014*; *Laranjinha et al., 2016*; *Cameron, 2016*). However, an autopsy study recently found the absence of micro-albuminuria in some patients with biopsy-proven DN (*Klessens et al., 2016*), which reminds us that a relative part of DN was clinically under-diagnosed. Furthermore, other studies of renal biopsy in DM have documented that non-diabetic renal disease (NDRD) plays a significant role in diabetic patients. The reported proportion of NDRD alone ranges from 18.1%–82.9% (*Pavkov et al., 2012*; *Byun et al., 2013*; *Sharma et al., 2013*; *Zhuo et al., 2013*; *Horvatic et al., 2014*; *Zwi et al., 2014*; *Klessens et al., 2016*; *Laranjinha et al., 2016*; *Liu et al., 2016*; *Liu, Tian & Jian, 2016*) in diabetic patients with renal disease, while that of NDRD superimposed on DN ranges from 7.8%–48.9% (*Oh et al., 2012*; *Pavkov et al., 2012*; *Byun et al., 2013*; *Zhuo et al., 2013*; *Horvatic et al., 2014*; *Zwi et al., 2014*; *Klessens et al., 2016*; *Laranjinha et al., 2016*; *Liu et al., 2016*; *Liu, Tian & Jian, 2016*). It is suggested that the absence of histopathologic evaluation may lead to a relatively high proportion of NDRD that were misdiagnosed as DN among diabetic patients. Thus, it is extremely important to further understand the spectrum of renal diseases in patients with DM, which leads to different therapies and prognosis.

However, most of the previous studies are from Western countries, where the spectrum of renal disease is different from that in China. There are only a few small-sampled, single-centered studies explaining the clinicopathological characteristics of DM related kidney disease in China (*Zhuo et al., 2013*; *Liu et al., 2016*; *Liu, Tian & Jian, 2016*). To date, comprehensive data on renal biopsy findings in Chinese diabetic patients are lacking. We have previously evaluated the profiles and temporal change of glomerular diseases in an 11-year renal biopsy series from 928 hospitals in 282 cities across China (*Xu et al., 2016*). In the current analysis, we aim to assess the modern spectrum of biopsy-proven renal disease in diabetic patients and analyze the clinical-pathological correlations, which could

remind physicians of large amout of NDRDs in diabetic patients and help them make better management to diabetic patients with renal disease.

## MATERIALS & METHODS

### Study population and clinical parameters

We previously conducted a nationwide renal biopsy survey over an 11-year period from January 2004 to December 2014. Among 71,151 native renal biopies, patients without histologic diagnosis or with less than five glomeruli under light microscopy, those with repeated biopsies, kidney graft, those with missing demographic or clinical data, and those diagnosed as isolated tubulointerstitial renal diseases were excluded from our study. Diabetic patients were identified according to the clinical diagnosis from referral records.

The referral record including the demographic and clinical data for each patient, who underwent kidney biopsy, was initially completed by nephrologists in local hospitals and subsequently sent to the central pathologic laboratories with the biopsy sample. Data on the following demographic and clinical variables were extracted from referral records and pathological reports of renal biopsies: age, gender, city of residence, date and hospital performing the biopsy, clinical diagnosis, indications for renal biopsy, serum creatinine (SCr), and histological diagnosis. eGFR (ml/min/1.73 m$^2$) was calculated from serum creatinine level using the CKD-EPI Creatinine Equation (2009) (*Levey et al., 2009*). The indications for renal biopsy included nephrotic syndrome (NS), acute kidney injury (AKI), chronic progressive kidney Injury (CPKI, defined as eGFR <60 ml/min/1.73 m$^2$), proteinuria without NS, isolated hematuria, and proteinuria coexisting with hematuria. All study patients were divided into three groups based on the biopsy findings: isolated DN group (DN), isolated NDRD group (NDRD), and NDRD superimposed on DN group (NDRD + DN).

The data from all study centers were pooled and analyzed at the National Clinical Research Center for Kidney Disease in Guangzhou. The Medical Ethics Committee of Nanfang Hospital, Southern Medical University approved the study protocol and waived patient consent; the approval number is NFEC-2015-073.

### Pathological diagnosis

The biopsies were conducted in local hospitals, as well as the histological specimen fixation. Samples were then processed at one of the six central pathologic laboratories and were diagnosed by one of the six leading pathologists. All the specimens were subjected to light microscopy examination (LM) and immunofluorescent staining, while 76.5% of biopsies had electron microscope (EM) examination. For light microscopy, hematoxylin and eosin (HE) staining, periodic acid Schiff's reagent (PAS) staining, periodic Schiff-methenamine (PASM) staining, and Masson's trichrome solution (Masson) staining were performed. In certain cases, Congo red and methyl violet staining were also done. Immunofluorescent staining for IgA, IgG, IgM, C3, C4, C1q, and κ/λ light chains was conducted. The histological findings were classified according to the "Revised Protocol for the Histological Typing of Glomerulopathy" (*Churg, Bernstein & Glassock, 1995*). The histological results were interpreted by the leading histopathologists and extracted from

electronic pathological reports. Histological diagnosis were made according to an uniform diagnostic criteria in six central pathological laboratories. There has been consistency in both the pathologic procedures and interpretations of biopsy specimens in the pathologic centers in charge of histologic diagnosis, and there were no substantial changes in the diagnosis of DN or differentiation of DN from other glomerular disease over the study period. DN was diagnosed based on the presence of diffuse or nodular glomerulosclerosis, mesangial (nodular or diffuse) widening, glomerular hypertrophy, glomerular capillary wall thickening, evidence of exudative lesions including fibrin caps, capsular drops, or hyaline thrombi. We reorganized the diagnoses based upon the 2010 DN classification: Class I for Mild or nonspecific LM changes and EM-proven glomerular basement membrane (GBM) thickening, Class II for mild to severe mesangial expansion, Class III for nodular sclerosis and Class IV for advanced diabetic glomerulosclerosis (*Tervaert et al., 2010*).

## Statistical analysis

All study data were stored in a standard Excel database. Statistical analysis was performed using SPSS version 20.0 for Windows (SPSS Inc., Chicago, IL, USA). Quantitative data was expressed as mean $\pm$ standard deviation (SD); categorical data was presented as frequencies and percentages ($n(\%)$). Differences between groups were analyzed by $\chi 2$ test or Fisher exact tests, if appropriate, and analysis of variance (ANOVA). The value of $P < 0.05$ was considered statistically significant.

## RESULTS

Among 71,151 patients who underwent renal biopsy, a total of 1,604 patients diagnosed as DM, which included 61 patients with type 1 diabetes mellitus (T1DM), were included and subsequently analyzed in the current study. The clinical characteristics of study population stratified by DN, NDRD or NDRD + DN, were summarized in Table 1. Out of the 1,604 patients, 717 patients (44.7%) had the isolated pathological diagnosis of DN, 788 (49.1%) had NDRD alone, and 99 (6.2%) had NDRD superimposed on DN. The median age is $51.39 \pm 11.37$ years. Patients in DN group were observed to be significantly younger than those in NDRD ($P = 0.001$). Male patients accounted for approximately 58% ($n = 932$) of the study population. Data on SCr were missing for 162 patients. Among the 1,442 cases with known SCr, patients in NDRD group had significantly lower SCr ($P < 0.001$) and higher eGFR ($P < 0.001$) than those in DN group. There was a rising trend in the number of diabetic patients who underwent renal biopsy over the study period, while the distribution of biopsy indications, patients' SCr level, and rate of NDRD remained stable. Patients diagnosed as DN were reorganized based upon the 2010 pathological classification of diabetic nephropathy: Class I–II for GBM thickening or mesangial expansion (65, 9.0%), Class III for nodular sclerosis (526, 73.4%) and Class IV for advanced diabetic glomerulosclerosis (126, 17.6%).

NS was the most common clinical indication for renal biopsy in all the three groups, and there was no difference in the incidence of nephrotic range proteinuria between the three groups ($P = 0.616$). Patients with DN had significantly lower incidence of AKI than patients with NDRD ($P < 0.001$). The proportion of CPKI was significantly lower in

**Table 1 Demographic and clinical characteristics of study population.** Shows the different demographic and clinical characteristics between patients with DN, patients with NDRD and patients with NDRD + DN.

| | DN n = 717 | NDRD n = 788 | NDRD + DN n = 99 | P value |
|---|---|---|---|---|
| Age | 50.24 ± 10.87 | 52.11 ± 11.89[a] | 53.82 ± 10.00[b] | 0.001 |
| Male sex | 433 (60.39%) | 434 (55.08%)[a] | 65 (65.66%)[c] | 0.033 |
| SCr | 145.37 ± 111.72 | 103.40 ± 102.80[a] | 171.28 ± 169.88[c] | <0.001 |
| eGFR | 62.21 ± 33.20 | 83.56 ± 31.96[a] | 61.83 ± 35.85[c] | <0.001 |
| Clinical indication | | | | |
| NS | 371 (51.7%) | 398 (50.5%) | 55 (55.6%) | 0.616 |
| AKI | 2 (0.3%) | 24 (3.0%)[a] | 5 (5.1%)[b] | <0.001 |
| CPKI | 105 (14.6%) | 50 (6.3%)[a] | 14 (14.1%)[c] | <0.001 |
| Proteinuria + hematuria | 97 (13.5%) | 151 (19.2%)[a] | 10 (10.1%)[b,c] | 0.003 |
| Proteinuria | 141 (19.7%) | 155 (19.7%) | 15 (15.2%) | 0.553 |
| Hematuria | 1 (0.1%) | 10 (1.3%)[a] | 0[b,c] | 0.024 |
| Time duration | | | | |
| 2004–2007 | 20 | 15 | 0 | |
| 2008–2011 | 108 | 141 | 11 | |
| 2012–2014 | 589 | 632 | 88 | |

**Notes.**

Quantitative data was expressed as mean ± standard deviation (SD), categorical data was presented as frequencies and percentages (n(%)). Differences between categorical data were analyzed by χ2 test; differences between quantitative data were analyzed by analysis of variance(ANOVA).

SCr, serum creatinine; eGFR, estimated glomerular filtration rate; T1DM, type 1 diabetes mellitus; NS, nephrotic syndrome; AKI, acute kidney injury; CKD, chronic kidney disease; DN, diabetic nephropahty; NDRD, non-diabetic renal disease.

[a] $P < 0.05$ for comparison of NDRD versus DN groups.

[b] $P < 0.05$ for comparison of NDRD + DN versus DN groups.

[c] $P < 0.05$ for comparison of NDRD + DN versus NDRD groups.

NDRD group than that in DN group and NDRD+DN group (6.3% vs 14.6%, 6.3% vs 14.1%, $P < 0.001$), whereas the proportion of proteinuria plus hematuria was significantly higher in NDRD than the other two groups ($P = 0.003$). Only one patient with DN presented with hematuria. Moreover, there was no significant difference between the incidence of subnephrotic proteinuria between the three groups.

Among 887 patients with NDRD, 788 patients had NDRD alone and 99 patients had NDRD superimposed on DN. The specific pathological diagnoses of NDRD were illustrated in Table 2. Membraneous nephropathy (MN) was the most common NDRD (357, 38.7%), followed by IgA nephropathy (IgAN) (179, 19.4%). Except for that, minimal change disease (MCD) (110, 11.9%), mesangial proliferative glomerulonephritis (MsPGN) (67, 7.3%), focal segmental glomerulosclerosis (FSGS) (47, 5.1%) remained the leading diagnoses of NDRD. Hypertensive renal disease ($n = 32$), tubulointerstitital nephropathy ($n = 27$) and acute tubular necrosis ($n = 16$) accounted for 3.5%, 2.9% and 1.7% of the NDRD cases respectively.

Additionally, a small number of patients (61, 3.8%) were diagnosed as T1DM. Among these population, 31 patients were identified as isolated DN, 29 patients were diagnosed

**Table 2  The spectrum of biopsy-proven renal disease in patients with NDRD.** The table shows the incidence of specific pathological diagnosis in patients with NDRD and NDRD + DN.

| | NDRD | NDRD + DN |
|---|---|---|
| Membraneous nephropathy | 323 (39.3%) | 34 (33.7%) |
| IgA nephropathy | 147 (17.9%) | 32 (31.7%)[a] |
| Minimal change disease | 110 (13.4%) | 0[a] |
| Mesangial proliferative glomerulonephritis | 67 (8.2%) | 0[a] |
| Focal segmental glomerulosclerosis | 46 (5.6%) | 1 (1.0%)[a] |
| Acute tubular necrosis | 15 (1.8%) | 1 (1.0%) |
| Tubulointerstitial nephropathy | 15 (1.8%) | 12 (11.9%)[a] |
| Hypertensive renal disease | 14 (1.7%) | 18 (17.8%)[a] |
| HBV-associated nephritis | 13 (1.6%) | 0 |
| Lupus nephritis | 12 (1.5%) | 0 |
| Henoch-Schonlein Purpura nephritis | 12 (1.5%) | 1 (1.0%) |
| ANCA-associated vasculitis | 7 (0.9%) | 0 |
| Proliferative sclerotic glomerulonephritis | 5 (0.6%) | 0 |
| Membraneous proliferative glomerulonephritis | 4 (0.5%) | 0 |
| Obesity associated nephropathy | 4 (0.5%) | 0 |
| Thin basement membrane nephropathy | 4 (0.5%) | 0 |
| Endocapillary proliferative glomerulonephritis | 2 (0.2%) | 0 |
| Amyloidosis-AL subtype | 2 (0.2%) | 0 |
| Amyloidosis | 2 (0.2%) | 0 |
| Mesangial nodular nephropathy | 2 (0.2%) | 0 |
| Focal glomerulonephritis | 1 (0.1%) | 0 |
| Crescentic glomerulonephritis | 1 (0.1%) | 0 |
| Immunotactoid glomerulopathy | 1 (0.1%) | 0 |
| Microscopic polyangiitis | 1 (0.1%) | 0 |
| Eosinophilic granulomatosis with polyangiitis | 1 (0.1%) | 0 |
| Light chain deposition disease | 1 (0.1%) | 0 |
| Fibrillary glomerulopathy | 0 | 2 (2.0%)[a] |
| Others | 10 (1.2%) | 0 |
| Total | 822 | 101 |

**Notes.**
Values are expressed as frequencies and percentages ($n$(%)).
NDRD, non-diabetic renal disease; DN, diabetic nephropathy.
Differences were analyzed by $\chi^2$ test or Fisher's Exact tests, if appropriate.
[a]$P < 0.05$ for comparison of NDRD + DN versus NDRD.

as pure NDRD, and one patient in the NDRD + DN group. IgAN (9, 30.0%) and MN (7, 23.3%) were the most common non-diabetic renal diseases in patients with T1DM.

## DISCUSSION

This is the largest, multi-centered study of renal biopsy findings in patients with DM in China. Among 1,604 diabetic patients, 788 patients (49.1%) were recognized as pure NDRD based on pathological features of renal biopsy, which accounts for nearly a half of the entire study population, while 717 patients (44.7%) had pure DN, and the remaining 99 (6.2%)

**Table 3 Summary of previous studies from different regions about NDRD in diabetic patients.** The table shows different spectrum of NDRD in diabetic patients among other regions during 2011–2016.

| | Duration | Number of patients | DN | NDRD | NDRD + DN | Most common NDRD |
|---|---|---|---|---|---|---|
| 2011 Korea | 1988–2008 | 119 T2DM | 43 (36.1%) | 64 (53.8%) | 12 (10.1%) | MN |
| 2012 Malaysia | 2004–2008 | 110 T2DM | 69 (62.7%) | 20 (18.2%) | 21 (19.1%) | AIN |
| 2013 China | 2003–2010 | 216 T2DM | 14 (6.5%) | 179 (82.9%) | 23 (10.7%) | IgAN |
| 2013 China | 2003–2011 | 244 T2DM | 20 (8.2%) | 205 (84%) | 19 (7.8%) | IgAN |
| 2013 Korea | 2000–2011 | 110 T2DM | 41 (37.3%) | 59 (53.6%) | 10 (9.1%) | IgAN |
| 2013 USA | 2011 | 620 T2DM | 227 (37%) | 220 (36%) | 164 (27%) | FSGS |
| 2014 Croatia | 2004–2013 | 80 T2DM | 37 (46.25%) | 29 (36.25%) | 14 (17.5%) | MN |
| 2014 New Zealand | 2004–2006 | 93 T2DM | 30 (32%) | 17 (18%) | 46 (49%) | FSGS |
| 2016 China | 2000–2015 | 273 T2DM | 68 (24.9%) | 175 (64.1%) | 30 (11.0%) | MN |

**Notes.**

Values are expressed as frequencies and percentages ($n$(%)).

NDRD, non-diabetic renal disease; DN, diabetic nephropathy; MN, membraneous nephropathy; AIN, acute interstitial nephritis; IgAN, IgA nephropathy; FSGS, focal segmental glomerulosclerosis.

patients had NDRD combined with DN. Our results provide an important supplement, as well as extension, to previous single-centered studies carried out in China with respect to renal biopsy in diabetic patients and better elucidate the characteristics of NDRD among diabetic patients in China.

A number of pathological diagnoses were identified in diabetic patients. The most common NDRD was MN, followed by IgAN in patients with DM, which is consistent with the high incidence of MN in China (*Xu et al., 2016*). However, previous studies found that IgAN was the leading type of NDRD in China (*Chong et al., 2012*; *Zhuo et al., 2013*). This difference may be attributed to the small sample size of previous researches and the changing pattern of glomerular disease in China during the last decade. The frequency of MN doubled from 2004 (12.2%) to 2014 (24.9%) in all renal biopsies, whereas the proportions of other major glomerulopathies remained stable. The composition of NDRD among diabetic patients varies across race and country. Comparison between studies from different countries since 2010 was presented in Table 3. MN and IgAN were the most common two pathological diagnoses in diabetic patients in the Asian population (*Tone et al., 2005*; *Oh et al., 2012*; *Byun et al., 2013*; *Liu et al., 2016*), as well as in Croatian (*Horvatic et al., 2014*). FSGS was much more common in the developed countries, such as the USA and New Zealand (*Sharma et al., 2013*; *Zwi et al., 2014*). The spectrum of NDRD in diabetic patients is consistent with the spectrum of glomerulonephritis in the same areas mentioned above. Additionally, AIN was the leading cause of NDRD in Malaysia and India (*Soni et al., 2006*; *Chong et al., 2012*). *Sharma et al. (2013)* found a high incidence of acute tubular necrnosis (ATN) in diabetic patients, while in our study, there are only 16 patients diagnosed as ATN, and all of them were complicated by other NDRD. It is probably due to the low proportion of patients with AKI in our study and regional variation of biopsy indications for patients with AKI. The most important aspect of this study may be that the spectrum of NDRD was identified in the majority of Chinese diabetic patients, which would yield significant changes in treatment, including the use of immunosuppressant.

According to previous researches, patients with T1DM presenting with renal impairment were more likely attributed to DN, which was quite different from pathological patterns observed in patients with T2DM. However, in our study, nearly a half (49.2%) of patients with T1DM had NDRD. IgAN and MN were the leading causes of NDRD in patients with T1DM, which was consistent with the composition and secular pattern of glomerulopathy in China. Due to the limited number of patients with T1DM in our study, more data are required to illustrate the precise spectrum of pathological diagnoses in patients with T1DM who had renal impairment.

There are some clinical parameters, such as short duration of DM, sudden onset of proteinuria, absence of diabetic retinopathy (DR), and presence of glomerular hematuria, observed to be useful in distinguishing NDRD from DN (*Lee, Chung & Choi, 1999*; *Liang et al., 2013*; *Sharma et al., 2013*; *Horvatic et al., 2014*; *Liu et al., 2014*; *Teng et al., 2014*; *Dong et al., 2016*). Several studies have previously shown that DM duration exceeding 10 years was a predictor of DN in diabetic patients (*Chang et al., 2011*; *Chong et al., 2012*; *Sharma et al., 2013*). However, there is no national registration system of DM in China, and the awareness rate of diabetes is extremely lower in China than that in western countries (*Li et al., 2013*; *Qin et al., 2016*). Thus, the self-reported disease duration in Chinese diabetic patients might be inaccurate, which cannot be an useful predictor for distinguishing DN from NDRD. Previous studies have suggested that massive proteinuria, especially the proteinuria in a nephrotic range, may be a predictive phenomenon for DN (*Gambara et al., 1993*; *Soni et al., 2006*; *Sharma et al., 2013*), while other recent studies showed no differences (*Liang et al., 2013*; *Horvatic et al., 2014*; *Liu et al., 2016*). In our study, nephrotic syndrome was the most common indication for biopsy in all diabetic patients. There are no significant differences of nephrotic range proteinuria between DN and NDRD group, as well as subnephrotic range proteinuria. An autopsy study also found that nearly 20% of histologically proven DN patients did not present with DN-associated clinical manifestations, such as proteinuria or diabetic retinopathy within their lifetime (*Klessens et al., 2016*). It suggested that DN may develop before the onset of clinical features. Although DR was considered strongly associated with DN (*Bergner et al., 2006*; *Pham et al., 2007*), one study suggested that DR is only associated with albuminuria DN, not normoalbuminuria DN (*Sabanayagam et al., 2014*). Another study from India demonstrated DR is also a poor predictor of DN in proteinuric diabetic patients (*Prakash et al., 2015*). However, patients with NDRD had a significantly higher rate of hematuria (with or without proteinuria) than those in DN group (19.2% vs 13.5%, 1.3% vs 0.1%), which indicates that hematuria may play a crucial part to forecast NDRD in diabetic patients. Recent study has also clarified that dysmorphic erythrocyte may predict the presence of NDRD in diabetic patients (*Dong et al., 2016*), while more data are required to discover the correlation between hematuria and NDRD in diabetic patients. In general, it still remains difficult to differentiate DN from NDRD in the clinical settings without the aid of renal biopsy.

The strength of this study mainly depends on the large number of patients and broad coverage of China. Because it is the first multi-centered study that performed in extensive areas of China, these results may illustrate the current national spectrum of biopsy-proven renal disease in diabetic patients. Only 44.7% of all the diabetic patients undergoing renal

biopsy were proved to be DN alone. Among the remaining 55.3% of patients, whether with pure NDRD or NDRD superimposed on DN, the MN and IgAN were the most common causative pathological types, which reminds us of that primary GN is still a major problem complicating the diabetic patients in China.

Our study still has some limitations. First of all, the selection bias is inevitable. Our patients were recruited by different nephrologists working in different regions of China. The indications of renal biopsy among diabetic patients may be not consistent with those recommended (including short duration of DM, sudden-onset decline in renal function and absence of DR *Dhaun et al., 2014*; *Teng et al., 2014*). Secondly, pathological diagnosis identified by one of the six leading pathologists could unavoidably introduce reporting bias to the results. However, uniform diagnostic criteria and consistency in pathologic procedure and interpretation in the central pathological laboratories would make up for this bias. Furthermore, a part of the clinical and laboratory data was not available in our study, such as duration of DM, presence or absence of DR, and 24-hours quantitative measurement of proteinuria, which is a major obstacle for us to further explore the clinical-pathological correlations in diabetic patients.

## CONCLUSIONS

To sum up, our results demonstrate that over 55% diabetic patient with kidney disease are diagnosed as NDRD either alone or coexisted with DN, among which MN and IgAN are the most common two pathological types. Patients with NDRD alone are more likely to have older age and lower serum creatinine level. CPKI are more prevalent in isolated DN patients, While hematuria seems more prevalent in NDRD patients. Neither nephrotic range proteinuria nor subnephrotic proteinuria shows significant differences between groups. Given the large percentage of NDRD features identified through biopsies in our cohort of diabetic patients with renal disease, and given that there were very few differences in the clinical presentation between the DN and NDRD groups, the study strongly supports the need for early renal biopsy in diabetic patients which can affect the long term management of these patients.

**List of abbreviations**

| | |
|---|---|
| **ANOVA** | analysis of variance |
| **DN** | diabetic nephropathy |
| **NDRD** | non-diabetic renal disease |
| **NS** | nephrotic syndrome |
| **MN** | membraneous nephropathy |
| **IgAN** | IgA nephropathy |
| **MCD** | minimal change disease |
| **MsPGN** | mesangial proliferative glomerulonephritis |
| **FSGS** | focal segmental glomerulonephrosclerosis |
| **ATN** | acute tubular necrosis |
| **CKD** | chronic kidney disease |
| **ESRD** | end stage renal disease |

| | |
|---|---|
| **DM** | diabetes mellitus; |
| **eGFR** | estimated glomerular filtration rate |
| **T1DM** | type 1 diabetes mellitus |
| **SCr** | serum creatinine |
| **AKI** | acute kidney injury |
| **CPKI** | chronic progressive kidney injury |
| **LM** | light microscopy |
| **EM** | electron microscopy |
| **HE** | hematoxylin and eosin |
| **PAS** | periodic acid Schiff's reagent |
| **PASM** | periodic Schiff-methenamine |
| **GBM** | glomerular basement membrane |
| **DR** | diabetic retinopathy |
| **GN** | glomerulonephritis |

### Funding
The authors received no funding for this work.

### Competing Interests
The authors declare there are no competing interests.

### Author Contributions
- Diankun Liu performed the experiments, analyzed the data, contributed reagents/materials/analysis tools, prepared figures and/or tables, authored or reviewed drafts of the paper, approved the final draft, re-analysis the raw data and revision of the final manuscript.
- Ting Huang and Nan Chen performed the experiments, authored or reviewed drafts of the paper, approved the final draft.
- Gang Xu and Ping Zhang analyzed the data, authored or reviewed drafts of the paper, approved the final draft.
- Yang Luo performed the experiments, contributed reagents/materials/analysis tools, authored or reviewed drafts of the paper, approved the final draft.
- Yongping Wang analyzed the data, contributed reagents/materials/analysis tools, authored or reviewed drafts of the paper, approved the final draft.
- Tao Lu prepared figures and/or tables, authored or reviewed drafts of the paper, approved the final draft.
- Long Wang and Mengqi Xiong prepared figures and/or tables, authored or reviewed drafts of the paper, approved the final draft, re-analysis the raw data and revision of the final manuscript.
- Jian Geng conceived and designed the experiments, authored or reviewed drafts of the paper, approved the final draft.
- Sheng Nie conceived and designed the experiments, prepared figures and/or tables, authored or reviewed drafts of the paper, approved the final draft.

## Human Ethics

The following information was supplied relating to ethical approvals (i.e., approving body and any reference numbers):

The Medical Ethics Committee of Nanfang Hospital, Southern Medical University approved the study protocol and waived patient consent (approval number NFEC-2015-073).

## Data Availability

The raw data including all the diabetic patients selected from the larger population for our study is available as a Supplemental File and is also available at Figshare:

Liu, Diankun (2018): Raw data-PeerJ.xlsx. figshare. https://doi.org/10.6084/m9.figshare.5945560.v1.

## Supplemental Information

Supplemental information for this article can be found online at http://dx.doi.org/10.7717/peerj.4522#supplemental-information.

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
