# Peer review of "The modern spectrum of biopsy-proven renal disease in Chinese diabetic patients—a retrospective descriptive study"

_PeerJ, doi:10.7717/peerj.4522_

## Round 0.1 · original submission · Major Revisions

The article is a useful contribution to the literature on the spectrum of renal disease in diabetes, particularly in Asian subjects. However as the reviewers have noted some revision will be required before it can be accepted for publication. We look forward to seeing a revised version which takes the reviewers' comments into account

Reviewer 1 ·

Basic reporting

Thanks for asking me to review this manuscript.
In this manuscript, Dr Liu et al has performed a retrospective study to assess common histopathological findings in Chinese diabetic patients with kidney disease who underwent renal biopsy. Isolated diabetic nephropathy was only seen in 44.7% of the patients, while over 50% patients in this cohort were diagnosed as non-diabetic renal disease. This is an interesting study, which has a potential impact on further management of diabetic patients with kidney disease.
I have following comments for the authors to take into consideration.
1. It would be interesting to know any correlations between histopathological diagnosis based on renal biopsy and clinical parameters. For example, did more patients with NDRD have nephrotic range proteinuria than those with isolated diabetic nephropathy in this cohort? The authors briefly stated this in the discussion section. Can this issue be assessed in more details and presented in a separate paragraph in the result section? Then those results should be interpreted and discussed in the discussion section as well.
2. In the line 136, should be in the current study.
3. Please add P values in the first paragraph of the result section.
4. In the line 225, it seems that the sentence was not complete.
…..should be performed in diabetic patients with kidney disease?

Experimental design

no comment

Validity of the findings

no comments

Additional comments

Thanks for asking me to review this manuscript.
In this manuscript, Dr Liu et al has performed a retrospective study to assess common histopathological findings in Chinese diabetic patients with kidney disease who underwent renal biopsy. Isolated diabetic nephropathy was only seen in 44.7% of the patients, while over 50% patients in this cohort were diagnosed as non-diabetic renal disease. This is an interesting study, which has a potential impact on further management of diabetic patients with kidney disease.
I have following comments for the authors to take into consideration.
1. It would be interesting to know any correlations between histopathological diagnosis based on renal biopsy and clinical parameters. For example, did more patients with NDRD have nephrotic range proteinuria than those with isolated diabetic nephropathy in this cohort? The authors briefly stated this in the discussion section. Can this issue be assessed in more details and presented in a separate paragraph in the result section? Then those results should be interpreted and discussed in the discussion section as well.
2. In the line 136, should be in the current study.
3. Please add P values in the first paragraph of the result section.
4. In the line 225, it seems that the sentence was not complete.
…..should be performed in diabetic patients with kidney disease?

·

Basic reporting

The article is written in clear English maintaining the scientific and professional standards. A few typographical errors in tense and sentence structure were noted.
Line 101; coexisted – coexisting?
Line 150; base – based
Line 197; presented – presenting
Line 255; ends in mid sentence and needs to be completed

In two areas, the meaning of the sentence was unclear and need to be rephrased.
Lines 143 -145; NS was the most common clinical indication for renal biopsy in all the three groups and there was no significant difference between them – please indicate what differences the authors are referring to.
Lines 187,188; The spectrum of NDRD in diabetic patients is consistent with the spectrum of glomerulonephritis in the same area – please specify what is meant by the ‘same area’

Since there are a several abbreviations used, although they are described within the text, tabulating these abbreviations with their meanings at the end of the article will make it easier for the reader.

There is a reasonably good review of background literature and the data is compared with some of the previous findings.

The general structure of the articles and the format of the tables conform to the journal requirements.

The raw data is available as an excel file with comprehensive analysis of all the 71152 cases. It would be useful to include a legend that provides descriptions of the different column headers as many of them are in abbreviated form. Similarly, the label keys for numerical data entry (E.g 1 = yes; 2 = no) will make the raw data independently analyzable for future readers.

Experimental design

The study is essentially a descriptive study, although in lines 85 to 87, three aims have been mentioned; two of which allude to correlative analysis. The results and discussion do not contain much comparative analysis especially on the clinicopathological correlations and determining the usefulness of renal biopsy.

The research question for this study has not been clearly defined in this article. The study has predominantly followed a descriptive pattern and has identified differences in the patterns of nephropathy among the population studied.

The article introduces three aims in lines 85 to 87
1. Assess the modern spectrum of biopsy proven renal disease in diabetic patients
2. Analyse the clinical-pathological correlations
3. Determine the usefulness of renal biopsy in patients with DM

The first aim has been achieved as a descriptive analysis of the histopathological findings of the renal biopsies. The results mainly show that out of the 1604 diabetic patients, NDRD was identified in 887; including 99 in combination with NDRD and DN. These results are not very different to the results found in other studies as tabulated in Table 3.
The authors mention in lines 172-173, that this study provides an important supplement and an extension to previous single centered studies which is agreeable. However, in line 174 they mention that the study ‘better elucidates the characteristics of NDRD among diabetic patients in China’ which has not been adequately described in this article.

The second aim has been addressed to some extent in the results in reference to the SCr levels, presence of proteinuria plus haematuria and CKD. However, given the extensive availability of data, I feel a much deeper analysis and discussion should have been made on this aspect than what is currently available in the article.

It is not clear how the authors have attempted to address the third aim through this study as neither the raw data nor the analysis provides and insight into the usefulness of renal biopsy. This I feel has to be assessed in comparison to the clinical progression of patients or as a case control with biopsied vs non-biopsied DM patients and requires a different methodological approach than what could be achieved through this study. Therefore, this aim should either be removed from the article or better justified by the authors.

Although its not clearly mentioned in the article, the study appears to be of a retrospective nature. The sample population has been categorized into different groups based on previously made diagnoses and not on a strict case definition identified by the authors. The histopathological data have also been obtained through previous histopathological reporting. It does not appear that all the cases have undergone the same gamut of staining methods and microscopic analysis. These should also be included in the limitations mentioned in the last paragraph under discussion (lines 240-248)

The selection of cases and the protocols used for histopathological grading have been clearly mentioned. However, certain aspects in the study should be made clearer such as; were all cases subjected to LM. EM and IF?. What percentage of cases had special stains?

Validity of the findings

The study has provided some valuable information in terms of renal histopathological findings in diabetes patients which would have some clinical relevance in treatment. However as mentioned above, the results do not provide any major difference to what has been already described and how this study contributes to new knowledge should be better elaborated. The large number of cases covered through this study seems to be its only strength.

The conclusion section is relatively under-written. In line 253 and 254, the sentence ‘renal biopsy is the most accurate method compared with other clinical predictors to distinguish DN from NDRD’ is an unjustified statement since the study results or discussion do not provide any comparison of renal biopsy vs other clinical predictors. Therefore, this statement needs to be removed or clearly justified. The last sentence ends in mid sentence and presumably the section is incomplete

Additional comments

As mentioned above, the study predominantly forms a descriptive analysis of the renal histopathological reports made on renal biopsies collected through a previous survey. A large number of cases from a multicentered selection have been analysed.

The title should be refined to clearly indicate that the study is based on a retrospective analysis of histopathological reports as it does not appear that the slides have been directly assessed by the authors themselves.

Similarly, it should be clearly stated whether all tissue samples have undergone light microscopy, electron microscopy and immunofluorescence. If not, since the study involves a large sample, the authors could reselect the cases which have undergone all three modalities and reanalyze the data.

I feel these factors introduce significant reporting bias to the results which is a limitation that needs to be adequately addressed in the article.

Although the authors have stated three aims in the introduction of the article, all of these aims have not been adequately addressed in the subsequent analysis. The first aim has been addressed reasonably, the second aim to a lesser degree but given the extensive data available I feel this should have been discussed more elaborately. The third aim does not fall within the scope of this study and should be removed.

The conclusion section is weak and needs to be better written to clearly correlate with the outputs of this study and also to give a strong message on the clinical relevance of the findings.

In my opinion this study has excellent potential for a good scientific publication. However, major revisions are needed to enhance its scientific quality and academic integrity prior to publication.

---

## Round 0.2 · accepted · Accept

Thanks for revising your paper so thoroughly. Please note the comments made by the second reviewer and consider making the corrections suggested during the production process.

·

Basic reporting

I commend the authors for having revised the article well.

A few typographical errors in the revised document were noted. 
Line 55; capitalization of other after Furthermore,
Line 90; sbiopsy – biopsy
Line 151 & 157, the sentences “there was no significant difference between the incidence of…..” should really read “there was no difference in the incidence of ….. between the three groups.”

Experimental design

The concerns mentioned in my previous review have been satisfactorily addressed.

Validity of the findings

The conclusion section is much improved. Although I feel the final sentence can be made to have a much bigger impact.

Suggestion – Given the large percentage of NDRD features identified through biopsies in our cohort of Diabetic patients with renal disease, and given that there were very few differences in the clinical presentation between the DN and NDRD groups, the study strongly supports the need for early renal biopsy in diabetic patients which can affect the long term management of these patients.

Additional comments

Many of the suggestions and concerns have been adequately addressed in this revised version and the article is much improved. Please see the minor revisions suggested above.

I feel mentioning the title as a ‘retrospective case series’ might unnecessarily bring down the value of this article as this is more than a case series. It would be better to call it – a retrospective descriptive study. 

I feel the article is now in a sufficiently suitable standard for publication once these minor revisions are made.